# Non-parametric Structured Output Networks

**Andreas M. Lehrmann**
Disney Research
Pittsburgh, PA 15213
andreas.lehrmann@disneyresearch.com

**Leonid Sigal**
Disney Research
Pittsburgh, PA 15213
lsigal@disneyresearch.com

## Abstract

Deep neural networks (DNNs) and probabilistic graphical models (PGMs) are the two main tools for statistical modeling. While DNNs provide the ability to model rich and complex relationships between input and output variables, PGMs provide the ability to encode dependencies among the output variables themselves. End-to-end training methods for models with structured graphical dependencies on top of neural predictions have recently emerged as a principled way of combining these two paradigms. While these models have proven to be powerful in discriminative settings with discrete outputs, extensions to structured continuous spaces, as well as performing efficient inference in these spaces, are lacking. We propose non-parametric structured output networks (NSON), a modular approach that cleanly separates a non-parametric, structured posterior representation from a discriminative inference scheme but allows joint end-to-end training of both components. Our experiments evaluate the ability of NSONs to capture structured posterior densities (modeling) and to compute complex statistics of those densities (inference). We compare our model to output spaces of varying expressiveness and popular variational and sampling-based inference algorithms.

## 1 Introduction

In recent years, deep neural networks have led to tremendous progress in domains such as image classification [1, 2] and segmentation [3], object detection [4, 5] and natural language processing [6, 7]. These achievements can be attributed to their hierarchical feature representation, the development of effective regularization techniques [8, 9] and the availability of large amounts of training data [10, 11].

While a lot of effort has been spent on identifying optimal network structures and trainings schemes to enable these advances, the expressiveness of the output space has not evolved at the same rate. Indeed, it is striking that most neural architectures model *categorical* posterior distributions that *do not* incorporate any structural assumptions about the underlying task; they are discrete and global (Figure 1a). However, many tasks are naturally formulated as structured problems or would benefit from continuous representations due to their high cardinality. In those cases, it is desirable to learn an expressive posterior density reflecting the dependencies in the underlying task.

As a simple example, consider a stripe of $n$ noisy pixels in a natural image. If we want to learn a neural network that encodes the posterior distribution $p_{\boldsymbol{\theta}}(\mathbf{y} \mid \mathbf{x})$ of the clean output $\mathbf{y}$ given the noisy input $\mathbf{x}$, we must ensure that $p_{\boldsymbol{\theta}}$ is expressive enough to represent potentially complex noise distributions and structured enough to avoid modeling spurious dependencies between the variables.

Probabilistic graphical models [12], such as Bayesian networks or Markov random fields, have a long history in machine learning and provide principled frameworks for such structured data. It is therefore natural to use their factored representations as a means of enforcing structure in a deep neural network. While initial results along this line of research have been promising [13, 14], they focus exclusively on the discrete case and/or mean-field inference.

Instead, we propose a deep neural network that encodes a non-parametric posterior density that factorizes over a graph (Figure 1b). We perform recurrent inference inspired by message-passing in this structured output space and show how to learn all components end-to-end.

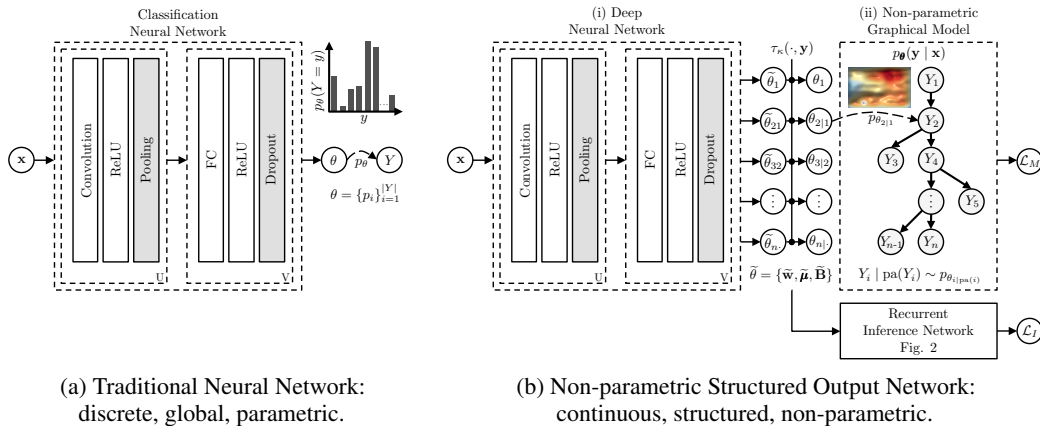

(a) Traditional Neural Network:
discrete, global, parametric.

(b) Non-parametric Structured Output Network:
continuous, structured, non-parametric.

Figure 1: **Overview: Non-parametric Structured Output Networks.** **(a)** Traditional neural networks use a series of convolution and inner product modules to predict a discrete posterior without graphical structure (*e.g.*, VGG [15]). [grey $\widehat{=}$ optional] **(b)** Non-parametric structured output networks use a deep neural network to predict a non-parametric graphical model $p_{\boldsymbol{\theta}(\mathbf{x})}(\mathbf{y})$ (NGM) that factorizes over a graph. A recurrent inference network (RIN) computes statistics $t[p_{\boldsymbol{\theta}(\mathbf{x})}(\mathbf{y})]$ from this structured output density. At training time, we propagate stochastic gradients from both NGM and RIN back to the inputs.

## 1.1 Related Work

Our framework builds upon elements from neural networks, structured models, non-parametric statistics, and approximate inference. We will first present prior work on structured neural networks and then discuss the relevant literature on approximate non-parametric inference.

### 1.1.1 Structured Neural Networks

Structured neural networks combine the expressive representations of deep neural networks with the structured dependencies of probabilistic graphical models. Early attempts to combine both frameworks used high-level features from neural networks (*e.g.*, `fc7`) to obtain fixed unary potentials for a graphical model [18]. More recently, statistical models and their associated inference tasks have been reinterpreted as (layers in) neural networks, which has allowed true end-to-end training and blurred the line between both paradigms: [13, 14] express the classic mean-field update equations as a series of layers in a recurrent neural network (RNN). Structure inference machines [17] use an RNN to simulate message-passing in a graphical model with soft-edges for activity recognition. A full backward-pass through loopy-BP was proposed in [19]. The structural-RNN [16] models all node and edge potentials in a spatio-temporal factor graph as RNNs that are shared among groups of nodes/edges with similar semantics. Table 1 summarizes some important properties of these methods. Notably, all output spaces except for the non-probabilistic work [16] are discrete.

### 1.1.2 Inference in Structured Neural Networks

In contrast to a discrete and global posterior, which allows inference of common statistics (*e.g.*, its mode) in linear time, expressive output spaces, as in Figure 1b, require message-passing schemes [20]

| Output Space | Related Work | | | | | |
|---|---|---|---|---|---|---|
| | VGG | MRF-RNN | Structural RNN | Structure Inference Machines | Deep Structured Models | NSON |
| | [15] | [14] | [16] | [17] | [13] | (**ours**) |
| Continuous | ✗ | ✗ | ✓ | ✗ | ✗ | ✓ |
| −Non-parametric | − | − | ✗ | − | − | ✓ |
| Structured | ✗ | ✓ | ✓ | ✓ | ✓ | ✓ |
| −End-to-end Training | − | ✓ | ✓ | ✗ | ✓ | ✓ |
| Prob. Inference | D | MF | ✗ | MP | MF | MP |
| Posterior Sampling | ✓ | ✗ | ✗ | ✓ | ✗ | ✓ |

Table 1: **Output Space Properties Across Models.**
[MF: mean-field; MP: message passing; D: direct; '−': not applicable]

to propagate and aggregate information. Local potentials outside of the exponential family, such as non-parametric distributions, lead to intractable message updates, so one needs to resort to approximate inference methods, which include the following two popular groups:

**Variational Inference.** Variational methods, such as mean-field and its structured variants [12], approximate an intractable target distribution with a tractable variational distribution by maximizing the evidence lower bound (ELBO). Stochastic extensions allow the use of this technique even on large datasets [21]. If the model is not in the conjugate-exponential family [22], as is the case for non-parametric graphical models, black box methods must be used to approximate an intractable expectation in the ELBO [23]. For fully-connected graphs with Gaussian pairwise potentials, the dense-CRF model [24] proposes an efficient way to perform the variational updates using the permutohedral lattice [25]. For general edge potentials, [26] proposes a density estimation technique that allows the use of non-parametric edge potentials.

**Sampling-based Inference.** This group of methods employs (sets of) samples to approximate intractable operations when computing message updates. Early works use iterative refinements of approximate clique potentials in junction trees [27]. Non-parametric belief propagation (NBP) [28, 29] represents each message as a kernel density estimate and uses Gibbs sampling for propagation. Particle belief propagation [30] represents each message as a set of samples drawn from an approximation to the receiving node's marginal, effectively circumventing the kernel smoothing required in NBP. Diverse particle selection [31] keeps a diverse set of hypothesized solutions at each node that pass through an iterative augmentation-update-selection scheme that preserves message values. Finally, a mean shift density approximation has been used as an alternative to sampling in [32].

## 1.2 Contributions

Our NSON model is inspired by the structured neural architectures (Section 1.1.1). However, in contrast to those approaches, we model structured dependencies on top of expressive non-parametric densities. In doing so, we build an inference network that computes statistics of these non-parametric output densities, thereby replacing the need for more conventional inference (Section 1.1.2).

In particular, we make the following contributions: (1) We propose non-parametric structured output networks, a novel approach combining the predictive power of deep neural networks with the structured representation and multimodal flexibility of non-parametric graphical models; (2) We show how to train the resulting output density together with recurrent inference modules in an end-to-end way; (3) We compare non-parametric structured output networks to a variety of alternative output densities and demonstrate superior performance of the inference module in comparison to variational and sampling-based approaches.

## 2 Non-parametric Structured Output Networks

Traditional neural networks (Figure 1a; [15]) encode a discrete posterior distribution by predicting an input-conditioned parameter vector $\widetilde{\theta}(\mathbf{x})$ of a categorical distribution, *i.e.*, $Y \mid X = \mathbf{x} \sim p_{\widetilde{\theta}(\mathbf{x})}$.

Non-parametric structured output networks (Figure 1b) do the same, except that $\widetilde{\boldsymbol{\theta}}(\mathbf{x})$ parameterizes a continuous graphical model with non-parametric potentials. It consists of three components: A deep neural network (DNN), a non-parametric graphical model (NGM), and a recurrent inference network (RIN). While the DNN+NGM encode a structured posterior ($\widehat{=}$ model), the RIN computes complex statistics in this output space ($\widehat{=}$ inference).

At a high level, the DNN, conditioned on an input $\mathbf{x}$, predicts the parameters $\widetilde{\boldsymbol{\theta}} = \{\widetilde{\theta}_{ij}\}$ (*e.g.*, kernel weights, centers and bandwidths) of local non-parametric distributions over a node and its parents according to the NGM's graph structure (Figure 1b). Using a function $\tau_\kappa$, these local joint distributions are then transformed to conditional distributions parameterized by $\boldsymbol{\theta} = \{\theta_{i|j}\}$ (*e.g.*, through a closed-form conditioning operation) and assembled into a structured joint density $p_{\boldsymbol{\theta}(\mathbf{x})}(\mathbf{y})$ with conditional (in)dependencies prescribed by the graphical model. Parameters of the DNN are optimized with respect to a maximum-likelihood loss $\mathcal{L}_M$. Simultaneously, a recurrent inference network (detailed in Figure 2) that takes $\widetilde{\boldsymbol{\theta}}$ as input, is trained to compute statistics of the structured distribution (*e.g.*, marginals) using a separate inference loss $\mathcal{L}_I$. The following two paragraphs discuss these elements in more detail.

**Model (DNN+NGM).** The DNN is parameterized by a weight vector $\boldsymbol{\lambda}_M$ and encodes a function from a generic input space $\mathcal{X}$ to a Cartesian parameter space $\Theta^n$,

$$\mathbf{x} \xmapsto{\boldsymbol{\lambda}_M} \widetilde{\boldsymbol{\theta}}(\mathbf{x}) = (\widetilde{\theta}_{i,\mathrm{pa}(i)}(\mathbf{x}))_{i=1}^n, \tag{1}$$

each of whose components models a joint kernel density $(Y_i, \mathrm{pa}(Y_i)) \sim p_{\widetilde{\theta}_{i,\mathrm{pa}(i)}(\mathbf{x})}$ and thus, implicitly, the local conditional distribution $Y_i \mid \mathrm{pa}(Y_i) \sim p_{\theta_{i|\mathrm{pa}(i)}(\mathbf{x})}$ of a non-parametric graphical model

$$p_{\boldsymbol{\theta}(\mathbf{x})}(\mathbf{y}) = \prod_{i=1}^n p_{\theta_{i|\mathrm{pa}(i)}(\mathbf{x})}(y_i \mid \mathrm{pa}(y_i)) \tag{2}$$

over a structured output space $\mathcal{Y}$ with directed, acyclic graph $G = (\mathcal{Y}, E)$. Here, $\mathrm{pa}(\cdot)$ denotes the set of parent nodes *w.r.t.* $G$, which we fix in advance based on prior knowledge or structure learning [12]. The conditional density of a node $Y = Y_i$ with parents $Y' = \mathrm{pa}(Y_i)$ and parameters $\theta = \theta_{i|\mathrm{pa}(i)}(\mathbf{x})$ is thus given by[1]

$$p_\theta(y \mid y') = \sum_{j=1}^N w^{(j)} \cdot |\mathbf{B}^{(j)}|^{-1} \kappa(\mathbf{B}^{(-j)}(y - \mu^{(j)})), \tag{3}$$

where the differentiable kernel $\kappa(u) = \prod_i q(u_i)$ is defined in terms of a symmetric, zero-mean density $q$ with positive variance and the conditional parameters $\theta = (\mathbf{w}, \boldsymbol{\mu}, \mathbf{B}) \in \Theta$ correspond to the full set of kernel weights, kernel centers, and kernel bandwidth matrices, respectively.[2] The functional relationship between $\theta$ and its joint counterpart $\widetilde{\theta} = \widetilde{\theta}_{i,\mathrm{pa}(i)}(\mathbf{x})$ is mediated through a kernel-dependent conditioning operation $\tau_\kappa(\widetilde{\theta}) = \tau_\kappa(\widetilde{\mathbf{w}}, \widetilde{\boldsymbol{\mu}}, \widetilde{\mathbf{B}}) = \theta$ and can be computed in closed-form for a wide range of kernels, including Gaussian, cosine, logistic and other kernels with sigmoid CDF. In particular, for block decompositions $\widetilde{\mathbf{B}}^{(j)} = \begin{pmatrix} \widetilde{\mathbf{B}}_y^{(j)} & 0 \\ 0 & \widetilde{\mathbf{B}}_{y'}^{(j)} \end{pmatrix}$ and $\widetilde{\mu}^{(j)} = \begin{pmatrix} \widetilde{\mu}_y^{(j)} \\ \widetilde{\mu}_{y'}^{(j)} \end{pmatrix}$, we obtain

$$\tau_\kappa(\widetilde{\theta}) = \theta = \begin{cases} w^{(j)} \propto \widetilde{w}^{(j)} \cdot |\widetilde{\mathbf{B}}_{y'}^{(j)}|^{-1} \kappa(\widetilde{\mathbf{B}}_{y'}^{(-j)}(y' - \widetilde{\mu}_{y'}^{(j)})), \\ \mu^{(j)} = \widetilde{\mu}_y^{(j)}, \qquad\qquad\qquad\qquad\qquad\quad 1 \le j \le N \\ \mathbf{B}^{(j)} = \widetilde{\mathbf{B}}_y^{(j)}. \end{cases} \tag{4}$$

See Appendix A.1 for a detailed derivation. We refer to the structured posterior density in Eq. (2) with the non-parametric local potentials in Eq. (3) as a *non-parametric structured output network*.

Given an output training set $\mathcal{D}_\mathcal{Y} = \{\mathbf{y}^{(i)} \in \mathcal{Y}\}_{i=1}^{N'}$, traditional kernel density estimation [33] can be viewed as an extreme special case of this architecture in which the discriminative, trainable DNN is replaced with a generative, closed-form estimator and $n := 1$ (no structure), $N := N'$ (#kernels = #training points), $w^{(i)} := (N')^{-1}$ (uniform weights), $\mathbf{B}^{(i)} := \mathbf{B}^{(0)}$ (shared covariance) and $\mu^{(i)} := \mathbf{y}^{(i)}$ (fixed centers). When learning $\boldsymbol{\lambda}_M$ from data, we can easily enforce parts or all of those restrictions in our model (see Section 5), but Section 3 will provide all necessary derivations for the more general case shown above.

**Inference (RIN).** In contrast to traditional classification networks with discrete label posterior, non-parametric structured output networks encode a complex density with rich statistics. We employ a recurrent inference network with parameters $\boldsymbol{\lambda}_I$ to compute such statistics $t$ from the predicted parameters $\widetilde{\boldsymbol{\theta}}(\mathbf{x}) \in \Theta^n$,

$$\widetilde{\boldsymbol{\theta}}(\mathbf{x}) \xmapsto{\boldsymbol{\lambda}_I} t[p_{\boldsymbol{\theta}(\mathbf{x})}]. \tag{5}$$

Similar to conditional graphical models, the underlying assumption is that the input-conditioned density $p_{\boldsymbol{\theta}(\mathbf{x})}$ contains all information about the semantic entities of interest and that we can infer whichever statistic we are interested in from it. A popular example of a statistic is a summary statistic,

$$t[p_{\boldsymbol{\theta}(\mathbf{x})}](y_i) = \mathrm{op}_{\mathbf{y}\setminus y_i} p_{\boldsymbol{\theta}(\mathbf{x})}(\mathbf{y}) \, \mathrm{d}(\mathbf{y}\setminus y_i), \tag{6}$$

which is known as sum-product BP ($\mathrm{op} = \int$; computing marginals) and max-product BP ($\mathrm{op} = \max$; computing max-marginals). Note, however, that we can attach recurrent inference networks corresponding to arbitrary tasks to this meta representation. Section 4 discusses the necessary details.

## 3 Learning Structured Densities using Non-Parametric Back-Propagation

The previous section introduced the model and inference components of a non-parametric structured output network. We will now describe how to learn the model (DNN+NGM) from a supervised training set $(\mathbf{x}^{(i)}, \mathbf{y}^{(i)}) \sim p_{\mathcal{D}}$.

### 3.1 Likelihood Loss

We write $\boldsymbol{\theta}(\mathbf{x}; \boldsymbol{\lambda}_M) = \tau_\kappa(\widetilde{\boldsymbol{\theta}}(\mathbf{x}; \boldsymbol{\lambda}_M))$ to explicitly refer to the weights $\boldsymbol{\lambda}_M$ of the deep neural network predicting the non-parametric graphical model (Eq. (1)). Since the parameters of $p_{\boldsymbol{\theta}(\mathbf{x})}$ are deterministic predictions from the input $\mathbf{x}$, the only free and learnable parameters are the components of $\boldsymbol{\lambda}_M$.

We train the DNN via empirical risk minimization with a negative log-likelihood loss $\mathcal{L}_M$,

$$\begin{aligned} \boldsymbol{\lambda}_M^* &= \underset{\boldsymbol{\lambda}_M}{\arg\min} \ \mathbb{E}_{(\mathbf{x},\mathbf{y}) \sim \widehat{p}_{\mathcal{D}}} [\mathcal{L}_M(\boldsymbol{\theta}(\mathbf{x}; \boldsymbol{\lambda}_M), \mathbf{y})] \\ &= \underset{\boldsymbol{\lambda}_M}{\arg\max} \ \mathbb{E}_{(\mathbf{x},\mathbf{y}) \sim \widehat{p}_{\mathcal{D}}} [\log p_{\boldsymbol{\theta}(\mathbf{x}; \boldsymbol{\lambda}_M)}(\mathbf{y})], \end{aligned} \tag{7}$$

where $\widehat{p}_{\mathcal{D}}$ refers to the empirical distribution and the expectation in Eq. (7) is taken over the factorization in Eq. (2) and the local distributions in Eq. (3). Note the similarities and differences between a non-parametric structured output network and a non-parametric graphical model with unary potentials from a neural network: Both model classes describe a structured posterior. However, while the unaries in the latter perform a reweighting of the potentials, a non-parametric structured output network predicts those potentials directly and allows joint optimization of its DNN and NGM components by back-propagating the structured loss first through the nodes of the graphical model and then through the layers of the neural network all the way back to the input.

### 3.2 Topological Non-parametric Gradients

We optimize Eq. (7) via stochastic gradient descent of the loss $\mathcal{L}_M$ *w.r.t.* the deep neural network weights $\boldsymbol{\lambda}_M$ using Adam [34]. Importantly, the gradients $\nabla_{\boldsymbol{\lambda}_M} \mathcal{L}_M(\boldsymbol{\theta}(\mathbf{x}; \boldsymbol{\lambda}_M), \mathbf{y})$ decompose into a factor from the deep neural network and a factor from the non-parametric graphical model,

$$\nabla_{\boldsymbol{\lambda}_M} \mathcal{L}_M(\boldsymbol{\theta}(\mathbf{x}; \boldsymbol{\lambda}_M), \mathbf{y}) = \frac{\partial \ \log p_{\boldsymbol{\theta}(\mathbf{x}; \boldsymbol{\lambda}_M)}(\mathbf{y})}{\partial \ \widetilde{\boldsymbol{\theta}}(\mathbf{x}; \boldsymbol{\lambda}_M)} \cdot \frac{\partial \ \widetilde{\boldsymbol{\theta}}(\mathbf{x}; \boldsymbol{\lambda}_M)}{\partial \ \boldsymbol{\lambda}_M}, \tag{8}$$

where the partial derivatives of the second factor can be obtained via standard back-propagation and the first factor decomposes according to the graphical model's graph structure $G$,

$$\frac{\partial \ \log p_{\boldsymbol{\theta}(\mathbf{x}; \boldsymbol{\lambda}_M)}(\mathbf{y})}{\partial \ \widetilde{\boldsymbol{\theta}}(\mathbf{x}; \boldsymbol{\lambda}_M)} = \sum_{i=1}^{n} \frac{\partial \ \log p_{\theta_{i|\mathrm{pa}(i)}(\mathbf{x}; \boldsymbol{\lambda}_M)}(y_i \mid \mathrm{pa}(y_i))}{\partial \ \widetilde{\boldsymbol{\theta}}(\mathbf{x}; \boldsymbol{\lambda}_M)}. \tag{9}$$

The gradient of a local model *w.r.t.* the joint parameters $\widetilde{\boldsymbol{\theta}}(\mathbf{x}; \boldsymbol{\lambda}_M)$ is given by two factors accounting for the gradient *w.r.t.* the conditional parameters and the Jacobian of the conditioning operation,

$$\frac{\partial \ \log p_{\theta_{i|\mathrm{pa}(i)}(\mathbf{x}; \boldsymbol{\lambda}_M)}(y_i \mid \mathrm{pa}(y_i))}{\partial \ \widetilde{\boldsymbol{\theta}}(\mathbf{x}; \boldsymbol{\lambda}_M)} = \frac{\partial \ \log p_{\theta_{i|\mathrm{pa}(i)}(\mathbf{x}; \boldsymbol{\lambda}_M)}(y_i \mid \mathrm{pa}(y_i))}{\partial \ \boldsymbol{\theta}(\mathbf{x}; \boldsymbol{\lambda}_M)} \cdot \frac{\partial \ \boldsymbol{\theta}(\mathbf{x}; \boldsymbol{\lambda}_M)}{\partial \ \widetilde{\boldsymbol{\theta}}(\mathbf{x}; \boldsymbol{\lambda}_M)}. \tag{10}$$

Note that the Jacobian takes a block-diagonal form, because $\theta = \theta_{i|\mathrm{pa}(i)}(\mathbf{x}; \boldsymbol{\lambda}_M)$ is independent of $\widetilde{\theta} = \widetilde{\theta}_{j,\mathrm{pa}(j)}(\mathbf{x}; \boldsymbol{\lambda}_M)$ for $i \neq j$. Each block constitutes the backward-pass through a node $Y_i$'s conditioning operation,

$$\frac{\partial \ \theta}{\partial \ \widetilde{\theta}} = \frac{\partial \ (\mathbf{w}, \boldsymbol{\mu}, \mathbf{B})}{\partial \ (\widetilde{\mathbf{w}}, \widetilde{\boldsymbol{\mu}}, \widetilde{\mathbf{B}})} = \begin{bmatrix} \frac{\partial \mathbf{w}}{\partial \widetilde{\mathbf{w}}} & \frac{\partial \mathbf{w}}{\partial \widetilde{\boldsymbol{\mu}}} & \frac{\partial \mathbf{w}}{\partial \widetilde{\mathbf{B}}} \\ \mathbf{0} & \frac{\partial \boldsymbol{\mu}}{\partial \widetilde{\boldsymbol{\mu}}} & \mathbf{0} \\ \mathbf{0} & \mathbf{0} & \frac{\partial \mathbf{B}}{\partial \widetilde{\mathbf{B}}} \end{bmatrix}, \tag{11}$$

where the individual entries are given by the derivatives of Eq. (4), *e.g.*,

$$\frac{\partial \mathbf{w}}{\partial \widetilde{\mathbf{w}}} = (\mathbf{w} \otimes \mathbf{w} + \mathrm{diag}(\mathbf{w})) \cdot \mathrm{diag}(\widetilde{\mathbf{w}})^{-1}. \tag{12}$$

Similar equations exist for the derivatives of the weights *w.r.t.* the kernel locations and kernel bandwidth matrices; the remaining cases are simple projections. In practice, we may be able to group the potentials $p_{\theta_{i|\mathrm{pa}(i)}}$ according to their semantic meaning, in which case we can train one potential per group instead of one potential per node by sharing the corresponding parameters in Eq. (9).

All topological operations can be implemented as separate layers in a deep neural network and the corresponding gradients can be obtained using automatic differentiation.

### 3.3 Distributional Non-parametric Gradients

We have shown how the gradient of the loss factorizes over the graph of the output space. Next, we will provide the gradients of those local factors $\log p_\theta(y \mid y')$ (Eq. (3)) *w.r.t.* the local parameters $\theta = \theta_{i|\mathrm{pa}(i)}$. To reduce notational clutter, we introduce the shorthand $\widehat{y}^{(k)} := \mathbf{B}^{(-k)}(y - \mu^{(k)})$ to refer to the normalized input and provide only final results; detailed derivations for all gradients and worked out examples for specific kernels can be found in Appendix A.2.

**Kernel Weights.**

$$\nabla_{\mathbf{w}} \log p_\theta(y \mid y') = \frac{\eta}{\mathbf{w}^\top \eta}, \quad \eta := \left( |\mathbf{B}^{(-k)}| \kappa(\widehat{y}^{(k)}) \right)_{k=1}^N. \tag{13}$$

Note that $\mathbf{w}$ is required to lie on the standard $(N-1)$-simplex $\Delta^{(N-1)}$. Different normalizations are possible, including a softmax or a projection onto the simplex, *i.e.*, $\pi_{\Delta^{(N-1)}}(w^{(i)}) = \max(0, w^{(i)} + u)$ and $u$ is the unique translation such that the positive points sum to 1 [35].

**Kernel Centers.**

$$\nabla_{\boldsymbol{\mu}} \log p_\theta(y \mid y') = \frac{\mathbf{w} \odot \beta}{\mathbf{w}^\top \eta}, \quad \beta := \left( \frac{-\mathbf{B}^{(-\top k)}}{|\mathbf{B}^{(k)}|} \cdot \frac{\partial \kappa(\widehat{y}^{(k)})}{\partial \widehat{y}^{(k)}} \right)_{k=1}^N. \tag{14}$$

The kernel centers do not underlie any spatial restrictions, but proper initialization is important. Typically, we use the centers of a $k$-means clustering with $k := N$ to initialize the kernel centers.

**Kernel Bandwidth Matrices.**

$$\nabla_{\mathbf{B}} \log p_\theta(y \mid y') = \frac{\mathbf{w} \odot \gamma}{\mathbf{w}^\top \eta}, \quad \gamma := \left( \frac{-\mathbf{B}^{(-\top k)}}{|\mathbf{B}^{(k)}|} \cdot \left( \kappa(\widehat{y}^{(k)}) + \frac{\partial \kappa(\widehat{y}^{(k)})}{\partial \widehat{y}^{(k)}} \widehat{y}^{(\top k)} \right) \right)_{k=1}^N. \tag{15}$$

While computation of the gradient *w.r.t.* $\mathbf{B}$ is a universal approach, specific kernels may allow alternative gradients: In a Gaussian kernel, for instance, the Gramian of the bandwidth matrix acts as a covariance matrix. We can thus optimize $\mathbf{B}^{(k)}\mathbf{B}^{(\top k)}$ in the interior of the cone of positive-semidefinite matrices by computing the gradients *w.r.t.* the Cholesky factor of the inverse covariance matrix.

## 4 Inferring Complex Statistics using Neural Belief Propagation

The previous sections introduced non-parametric structured output networks and showed how their components, DNN and NGM, can be learned from data. Since the resulting posterior density $p_{\boldsymbol{\theta}(\mathbf{x})}(\mathbf{y})$ (Eq. (2)) factorizes over a graph, we can, in theory, use local messages to propagate beliefs about statistics $t[p_{\boldsymbol{\theta}(\mathbf{x})}(\mathbf{y})]$ along its edges (BP; [20]). However, special care must be taken to handle intractable operations caused by non-parametric local potentials and to allow an end-to-end integration.

For ease of exposition, we assume that we can represent the local conditional distributions as a set of pairwise potentials $\{\phi(y_i, y_j)\}$, effectively converting our directed model to a normalized MRF. This is not limiting, as we can always convert a factor graph representation of Eq. (2) into an equivalent pairwise MRF [36]. In this setting, a BP message $\mu_{i \to j}(y_j)$ from $Y_i$ to $Y_j$ takes the form

$$\mu_{i \to j}(y_j) = \mathrm{op}_{y_i} \phi(y_i, y_j) \cdot \mu_{\cdot \to i}(y_i), \tag{16}$$

where the operator $\mathrm{op}_y$ computes a summary statistic, such as integration or maximization, and $\mu_{\cdot \to i}(y_i)$ is the product of all incoming messages at $Y_i$. In case of a graphical model with non-parametric local distributions (Eq. (3)), this computation is not feasible for two reasons: (1) the pre-messages $\mu_{\cdot \to i}(y_i)$ are products of sums, which means that the number of kernels grows exponentially in the number of incoming messages; (2) the functional $\mathrm{op}_y$ does not usually have an analytic form.

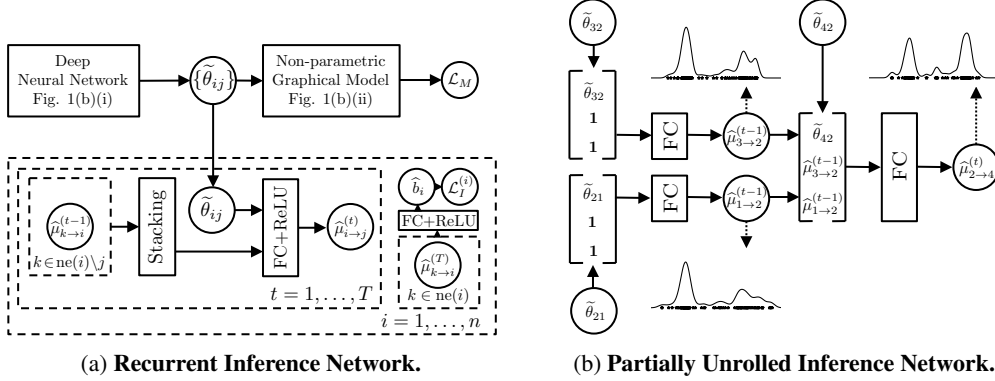

(a) **Recurrent Inference Network.**      (b) **Partially Unrolled Inference Network.**

Figure 2: **Inferring Complex Statistics.** Expressive output spaces require explicit inference procedures to obtain posterior statistics. We use an inference network inspired by message-passing schemes in non-parametric graphical models. **(a)** An RNN iteratively computes outgoing messages from incoming messages and the local potential. **(b)** Unrolled inference network illustrating the computation of $\widehat{\mu}_{2\to 4}$ in the graph shown in Figure 1b.

Inspired by recent results in imitation learning [37] and inference machines for classification [17, 38], we take an alternate route and use an RNN to model the exchange of information between non-parametric nodes. In particular, we introduce an RNN node $\widehat{\mu}_{i\to j}$ for each message and connect them in time according to Eq. (16), *i.e.*, each node has incoming connections from its local potential $\widetilde{\theta}_{ij}$, predicted by the DNN, and the nodes $\{\widehat{\mu}_{k\to i} : k \in \mathrm{ne}_G(i)\backslash j\}$, which correspond to the incoming messages. The message computation itself is approximated through an FC+ReLU layer with weights $\boldsymbol{\lambda}_I^{i\to j}$. An approximate message $\widehat{\mu}_{i\to j}$ from $Y_i$ to $Y_j$ can thus be written as

$$\widehat{\mu}_{i\to j} = \mathrm{ReLU}(\mathrm{FC}_{\boldsymbol{\lambda}_I^{i\to j}}(\mathrm{Stacking}(\widetilde{\theta}_{ij}, \{\widehat{\mu}_{k\to i} : k \in \mathrm{ne}_G(i)\backslash j\}))), \qquad (17)$$

where $\mathrm{ne}_G(\cdot)$ returns the neighbors of a node in $G$. The final beliefs $\widehat{b}_i = \widehat{\mu}_{\cdot\to i} \cdot \widehat{\mu}_{i\to j}$ can be implemented analogously. Similar to (loopy) belief updates in traditional message-passing, we run the RNN for a fixed number of iterations, at each step passing all neural messages. Furthermore, using the techniques discussed in Section 3.3, we can ensure that the messages are valid non-parametric distributions. All layers in this recurrent inference network are differentiable, so that we can propagate a decomposable inference loss $\mathcal{L}_I = \sum_{i=1}^n \mathcal{L}_I^{(i)}$ end-to-end back to the inputs. In practice, we find that generic loss functions work well (see Section 5) and that canonic loss functions can often be obtained directly from the statistic. The DNN weights $\boldsymbol{\lambda}_M$ are thus updated so as to do both predict the right posterior density and, together with the RIN weights $\boldsymbol{\lambda}_I$, perform correct inference in it (Figure 2).

## 5 Experiments

We validate non-parametric structured output networks at both the model (DNN+NGM) and the inference level (RIN). Model validation consists of a comparison to baselines along two binary axes, structured*ness* and non-parametr*icity*. Inference validation compares our RIN unit to the two predominant groups of approaches for inference in structured non-parametric densities, *i.e.*, sampling-based and variational inference (Section 1.1.2).

### 5.1 Dataset

We test our approach on simple natural pixel statistics from Microsoft COCO [11] by sampling stripes $\mathbf{y} = (y_i)_{i=1}^n \in [0, 255]^n$ of $n = 10$ pixels. Each pixel $y_i$ is corrupted by a linear noise model, leading to the observable output $x_i = \beta \cdot y_i + \epsilon$, with $\epsilon \sim \mathcal{N}(255 \cdot \delta_{\beta, -1}, \sigma^2)$ and $\beta \sim Ber(\psi)$, where the target space of the Bernoulli trial is $\{-1, +1\}$. For our experiments, we set $\sigma^2 = 100$ and $\psi = 0.5$. Using this noise process, we generate training and test sets of sizes 100,000 and 1,000, respectively.

### 5.2 Model Validation

The distributional gradients (Eq. (9)) comprise three types of parameters: Kernel locations, kernel weights, and kernel bandwidth matrices. Default values for the latter two exist in the form of uniform weights and plug-in bandwidth estimates [33], respectively, so we can turn optimization of those

| Model | Non-param. | Structured | Parameter Group Estimation | | | |
|---|---|---|---|---|---|---|
| | | | \-W | | +W | |
| | | | −B | +B | −B | +B |
| Gaussian | ✗ | ✗ | −1.13 (ML estimation) | | | |
| Kernel Density | ✓ | ✗ | +6.66 (Plug-in bandwidth estimation) | | | |
| Gaussian | ✗ | ✗ | −0.90 | +2.54 | −0.88 | +2.90 |
| GGM [39] | ✗ | ✓ | −0.85 | +1.55 | −0.93 | +1.53 |
| Mixture Density [40] | ✓ | ✗ | +9.22 | +6.87 | +11.18 | +11.51 |
| NGM-100 (ours) | ✓ | ✓ | +15.26 | +15.30 | +16.00 | **+16.46** |

(Last four rows labeled "Neural Network +")

(a) Model Validation

| Inference | Particles | Performance (marg. log-lik.) | Runtime (sec) |
|---|---|---|---|
| BB-VI [23] | 400 | +2.30 | 660.65 |
| | 800 | +3.03 | 1198.08 |
| P-BP [30] | 50 | +2.91 | 0.49 |
| | 100 | +6.13 | 2.11 |
| | 200 | +7.01 | 6.43 |
| | 400 | +8.85 | 21.13 |
| RIN-100 (ours) | – | **+16.62** | **0.04** |

(b) Inference Validation

Table 2: **Quantitative Evaluation. (a)** We report the expected log-likelihood of the test set under the predicted posterior $p_{\boldsymbol{\theta}(\mathbf{x})}(\mathbf{y})$, showing the need for a structured and non-parametric approach to model rich posteriors. **(b)** Inference using our RIN architecture is much faster than sampling-based or variational inference while still leading to accurate marginals. [(N/G)GM: Non-parametric/Gaussian Graphical Model; RIN-$x$: Recurrent Inference Network with $x$ kernels; P-BP: Particle Belief Propagation; BB-VI: Black Box Variational Inference]

parameter groups on/off as desired.[3] In addition to those variations, non-parametric structured output networks with a Gaussian kernel $\kappa = \mathcal{N}(\cdot \mid \vec{\mathbf{0}}, \mathbf{I})$ comprise a number of popular baselines as special cases, including neural networks predicting a Gaussian posterior ($n = 1, N = 1$), mixture density networks ($n = 1, N > 1$; [40]), and Gaussian graphical models ($n > 1, N = 1$; [39]). For the sake of completeness, we also report the performance of two basic posteriors without preceding neural network, namely a pure Gaussian and traditional kernel density estimation (KDE). We compare our approach to those baselines in terms of the expected log-likelihood on the test set, which is a relative measure for the KL-divergence to the true posterior.

**Setup and Results.** For the two basic models, we learn a joint density $p(\mathbf{y}, \mathbf{x})$ by maximum likelihood (Gaussian) and plug-in bandwidth estimation (KDE) and condition on the inputs $\mathbf{x}$ to infer the labels $\mathbf{y}$. We train the other 4 models for 40 epochs using a Gaussian kernel and a diagonal bandwidth matrix for the non-parametric models. The DNN consists of 2 fully-connected layers with 256 units and the kernel weights are constrained to lie on a simplex with a softmax layer. The NGM uses a chain-structured graph that connects each pixel to its immediate neighbors. Table 2a shows our results. Ablation study: unsurprisingly, a purely Gaussian posterior cannot represent the true posterior appropriately. A multimodal kernel density works better than a neural network with parametric posterior but cannot compete with the two non-parametric models attached to the neural network. Among the methods with a neural network, optimization of kernel locations only (first column) generally performs worst. However, the $-W + B$ setting (second column) gets sometimes trapped in local minima, especially in case of global mixture densities. If we decide to estimate a second parameter group, weights ($+W$) should therefore be preferred over bandwidths ($+B$). Best results are obtained when estimation is turned on for all three parameter groups. Baselines: the two non-parametric methods consistently perform better than the parametric approaches, confirming our claim that non-parametric densities are a powerful alternative to a parametric posterior. Furthermore, a comparison of the last two rows shows a substantial improvement due to our factored representation, demonstrating the importance of incorporating structure into high-dimensional, continuous estimation problems.

**Learned Graph Structures.** While the output variables in our experiments with one-dimensional pixel stripes have a canonical dependence structure, the optimal connectivity of the NGM in tasks with complex or no spatial semantics might be less obvious. As an example, we consider the case of two-dimensional image patches of size $10 \times 10$, which we extract and corrupt following the same protocol and noise process as above. Instead of specifying the graph by hand, we use a mutual information criterion [41] to learn the optimal arborescence from the training labels. With estimation of all parameter groups turned on ($+W + B$), we obtain results that are fully in line with those above: the expected test log-likelihood of NSONs ($+153.03$) is again superior to a global mixture density ($+76.34$), which in turn outperforms the two parametric approaches (GGM: $+18.60$; Gaussian: $−19.03$). A full ablation study as well as a visualization of the inferred graph structure are shown in Appendix A.3.

## 5.3 Inference Validation

Section 4 motivated the use of a recurrent inference network (RIN) to infer rich statistics from structured, non-parametric densities. We compare this choice to the other two groups of approaches, *i.e.*, variational and sampling-based inference (Section 1.1.2), in a marginal inference task. To this end, we pick one popular member from each group as baselines for our RIN architecture.

**Particle Belief Propagation (P-BP; [30]).** Sum-product particle belief propagation approximates a BP-message (Eq. (16); $\mathrm{op} := \int$) with a set of particles $\{y_j^{(s)}\}_{s=1}^S$ per node $Y_j$ by computing

$$\widehat{\mu}_{i \to j}(y_j^{(k)}) = \sum_{s=1}^{S} \frac{\phi(y_i^{(s)}, y_j^{(k)}) \cdot \widehat{\mu}_{\cdot \to i}(y_i^{(s)})}{S \rho(y_i^{(s)})}, \tag{18}$$

where the particles are sampled from a proposal distribution $\rho$ that approximates the true marginal by running MCMC on the beliefs $\widehat{\mu}_{\cdot \to i}(y_i) \cdot \widehat{\mu}_{i \to j}(y_i)$. Similar versions exist for other operators [42].

**Black Box Variational Inference (BB-VI; [23]).** Black box variational inference maximizes the ELBO $\mathcal{L}_{VI}[q_{\boldsymbol{\lambda}}]$ with respect to a variational distribution $q_{\boldsymbol{\lambda}}$ by approximating its gradient through a set of samples $\{\mathbf{y}^{(s)}\}_{s=1}^S \sim q_{\boldsymbol{\lambda}}$ and performing stochastic gradient ascent,

$$\nabla_{\boldsymbol{\lambda}} \mathcal{L}_{VI}[q_{\boldsymbol{\lambda}}] = \nabla_{\boldsymbol{\lambda}} \mathbb{E}_{q_{\boldsymbol{\lambda}}(\mathbf{y})} \left[ \log \frac{p_{\boldsymbol{\theta}}(\mathbf{y})}{q_{\boldsymbol{\lambda}}(\mathbf{y})} \right] \approx S^{-1} \sum_{s=1}^{S} \nabla_{\boldsymbol{\lambda}} \log q_{\boldsymbol{\lambda}}(\mathbf{y}^{(s)}) \log \frac{p_{\boldsymbol{\theta}}(\mathbf{y}^{(s)})}{q_{\boldsymbol{\lambda}}(\mathbf{y}^{(s)})}. \tag{19}$$

A statistic $t$ (Eq. (5)) can then be estimated from the tractable variational distribution $q_{\boldsymbol{\lambda}}(\mathbf{y})$ instead of the complex target distribution $p_{\boldsymbol{\theta}}(\mathbf{y})$. We use an isotropic Gaussian kernel $\kappa = \mathcal{N}(\cdot \mid \vec{\mathbf{0}}, \mathbf{I})$ together with the traditional factorization $q_{\boldsymbol{\lambda}}(\mathbf{y}) = \prod_{i=1}^{n} q_{\lambda_i}(y_i)$, in which case variational sampling is straighforward and the (now unconditional) gradients are given directly by Section 3.3.

### 5.3.1 Setup and Results.

We train our RIN architecture with a negative log-likelihood loss attached to each belief node, $\mathcal{L}_I^{(i)} = -\log p_{\theta_i}(y_i)$, and compare its performance to the results obtained from P-BP and BB-VI by calculating the sum of marginal log-likelihoods. For the baselines, we consider different numbers of particles, which affects both performance and speed. Additionally, for BB-VI we track the performance across 1024 optimization steps and report the best results. Table 2b summarizes our findings. Among the baselines, P-BP performs better than BB-VI once a required particle threshold is exceeded. We believe this is a manifestation of the special requirements associated with inference in non-parametric densities: while BB-VI needs to fit a high number of parameters, which poses the risk of getting trapped in local minima, P-BP relies solely on the evaluation of potentials. However, both methods are outperformed by a significant margin by our RIN, which we attribute to its end-to-end training in accordance with DNN+NGM and its ability to propagate and update full distributions instead of their mere value at a discrete set of points. In addition to pure performance, a key advantage of RIN inference over more traditional inference methods is its speed: our RIN approach is over $50\times$ faster than P-BP with 100 particles and orders of magnitude faster than BB-VI. This is significant, even when taking dependencies on hardware and implementation into account, and allows the use of expressive non-parametric posteriors in time-critical applications.

## 6 Conclusion

We proposed non-parametric structured output networks, a highly expressive framework consisting of a deep neural network predicting a non-parametric graphical model and a recurrent inference network computing statistics in this structured output space. We showed how all three components can be learned end-to-end by backpropagating non-parametric gradients through directed graphs and neural messages. Our experiments showed that non-parametric structured output networks are necessary for both effective learning of multimodal posteriors and efficient inference of complex statistics in them. We believe that NSONs are suitable for a variety of other structured tasks and can be used to obtain accurate approximations to many intractable statistics of non-parametric densities beyond (max-)marginals.

## Footnotes

[1]We write $\mathbf{B}^{(-j)} := \left(\mathbf{B}^{(j)}\right)^{-1}$ and $\mathbf{B}^{-T} := \left(\mathbf{B}^{-1}\right)^\top$ to avoid double superscripts.

[2]Note that $\theta$ represents the parameters of a specific node; different nodes may have different parameters.

[3]Since plug-in estimators depend on the kernel locations, the gradient *w.r.t.* the kernel locations needs to take these dependencies into account by backpropagating through the estimator and computing the total derivative.

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
