[Supplementary Material]

# Non-parametric Structured Output Networks (Supplementary Material)

## A  Appendix

### A.1  Forward-pass Through Non-parametric Conditional Distributions

As we did in the main paper, we fix a local variable $Y := Y_i$ with parents $Y' := \mathrm{pa}(Y_i)$ and parameterize the local conditional distribution $p_\theta(y \mid y')$ in terms of a parameter set $\theta = (w^{(j)}, \mu^{(j)}, \mathbf{B}^{(j)})_{j=1}^N$ that is related to the parameters $\widetilde{\theta} = (\widetilde{w}^{(j)}, \widetilde{\mu}^{(j)}, \widetilde{\mathbf{B}}^{(j)})_{j=1}^N$ of the corresponding joint distribution $p_{\widehat{\theta}}(y, y')$ through a $\kappa$-dependent conditioning operation $\tau_\kappa(\widetilde{\theta}) = \theta$. Writing $(\widehat{y}^{(i)}, \widehat{y}'^{(i)})^\top := \widetilde{\mathbf{B}}^{(-i)}\left((y, y')^\top - \widetilde{\mu}^{(i)}\right)$ for the normalized joint input[4] and assuming block decompositions $\widetilde{\mathbf{B}}^{(i)} = \begin{pmatrix} \widetilde{\mathbf{B}}_y^{(i)} & \mathbf{0} \\ \mathbf{0} & \widetilde{\mathbf{B}}_{y'}^{(i)} \end{pmatrix}$ and $\widetilde{\mu}^{(i)} = (\widetilde{\mu}_y^{(i)}, \widetilde{\mu}_{y'}^{(i)})^\top$ for the joint bandwidth matrices and joint kernels, respectively, we have

$$p_\theta(y \mid y') = \frac{\sum_{i=1}^N \widetilde{w}^{(i)} \cdot |\widetilde{\mathbf{B}}^{(i)}|^{-1} \kappa((\widehat{y}^{(i)}, \widehat{y}'^{(i)})^\top)}{\sum_{i=1}^N \widetilde{w}^{(i)} \cdot |\widetilde{\mathbf{B}}^{(i)}|^{-1} \int_y \kappa((\widehat{y}^{(i)}, \widehat{y}'^{(i)})^\top) \, \mathrm{d}y}. \tag{20}$$

After absorbing the evidence into the weights and factorizing the kernels, we obtain

$$p_\theta(y \mid y') = \sum_{i=1}^N \frac{\widetilde{w}^{(i)} \cdot |\widetilde{\mathbf{B}}^{(i)}|^{-1} \kappa(\widehat{y}'^{(i)})}{\sum_{j=1}^N \widetilde{w}^{(j)} \cdot |\widetilde{\mathbf{B}}^{(j)}|^{-1} \kappa(\widehat{y}'^{(j)}) \int_y \kappa(\widehat{y}^{(j)}) \, \mathrm{d}y} \kappa(\widehat{y}^{(i)}). \tag{21}$$

Using integration by substitution and the fact that $\kappa$ is a density, the integral evaluates to $\int_y \kappa(\widehat{y}^{(j)}) \, \mathrm{d}y = |\frac{\partial \, \widehat{y}^{(j)}}{\partial \, y}| = |\widetilde{\mathbf{B}}_y^{(j)}|$. The local conditional distribution can thus be expressed as

$$\begin{aligned}
p_\theta(y \mid y') &= \sum_{i=1}^N \frac{\widetilde{w}^{(i)} \cdot |\widetilde{\mathbf{B}}_{y'}^{(i)}|^{-1} \kappa(\widehat{y}'^{(i)})}{\sum_{j=1}^N \widetilde{w}^{(j)} \cdot |\widetilde{\mathbf{B}}_{y'}^{(j)}|^{-1} \kappa(\widehat{y}'^{(j)})} |\widetilde{\mathbf{B}}_y^{(i)}|^{-1} \kappa(\widehat{y}^{(i)}) \\
&= \sum_{i=1}^N w^{(i)} |\mathbf{B}^{(i)}|^{-1} \kappa(\mathbf{B}^{(-i)}(y - \mu^{(i)})),
\end{aligned} \tag{22}$$

where we have used the definitions

$$\begin{aligned}
w^{(i)} &:= \tau_\kappa(\widetilde{w}^{(i)}) = \frac{\widetilde{w}^{(i)} \cdot |\widetilde{\mathbf{B}}_{y'}^{(i)}|^{-1} \kappa(\widehat{y}'^{(i)})}{\sum_{j=1}^N \widetilde{w}^{(j)} \cdot |\widetilde{\mathbf{B}}_{y'}^{(j)}|^{-1} \kappa(\widehat{y}'^{(j)})}, \\
\mathbf{B}^{(i)} &:= \tau_\kappa(\widetilde{\mathbf{B}}^{(i)}) = \widetilde{\mathbf{B}}_y^{(i)}, \\
\mu^{(i)} &:= \tau_\kappa(\widetilde{\mu}^{(i)}) = \widetilde{\mu}_y^{(i)}.
\end{aligned} \tag{23}$$

### A.1.1  Forward-pass with Gaussian Kernels

As a concrete example, we derive the non-parametric conditional distribution Eq. (22) and its corresponding conditioning operation Eq. (23) for an isotropic Gaussian kernel $\kappa(\cdot) = \mathcal{N}(\cdot \mid \vec{\mathbf{0}}, \mathbf{I})$. Both results follow immediately from the observation $|\mathbf{B}^{(i)}|^{-1} \kappa(\mathbf{B}^{(-i)}(y - \mu^{(i)})) = \mathcal{N}(y \mid \mu^{(i)}, \mathbf{B}^{(2i)})$, leading to

$$p_\theta(y \mid y') = \sum_{i=1}^N w^{(i)} \mathcal{N}(y \mid \mu^{(i)}, \mathbf{B}^{(2i)}), \tag{24}$$

where the conditioning operation $\tau$ is given by

$$
\begin{aligned}
w^{(i)} = \tau_\kappa(\widetilde{w}^{(i)}) &= \frac{\widetilde{w}^{(i)}\mathcal{N}(y' \mid \widetilde{\mu}_{y'}^{(i)}, \widetilde{\mathbf{B}}_{y'}^{(2i)})}{\sum_{j=1}^{N} \widetilde{w}^{(j)}\mathcal{N}(y' \mid \widetilde{\mu}_{y'}^{(j)}, \widetilde{\mathbf{B}}_{y'}^{(2j)})}, \\
\mathbf{B}^{(i)} = \tau_\kappa(\widetilde{\mathbf{B}}^{(i)}) &= \widetilde{\mathbf{B}}_y^{(i)}, \\
\mu^{(i)} = \tau_\kappa(\widetilde{\mu}^{(i)}) &= \widetilde{\mu}_y^{(i)},
\end{aligned}
\tag{25}
$$

*i.e.*, a conditional Gaussian mixture whose weights are updated based on the likelihood of the evidence and whose means and covariances are given by the kernel locations and the Gramians of the bandwidth matrices, respectively.

## A.2 Backward-pass Through Non-parametric Conditional Distributions

We derive Eqs. (13)–(15) in the main paper in some more detail. Note that the results below only account for the gradients *w.r.t.* the conditional parameters and must be multiplied by the Jacobian of the conditioning operation to obtain the gradients *w.r.t.* the joint parameters (Eq. (10)).

First observe that for each mixture density $p(y) = \sum_{i=1}^{N} w^{(i)} \cdot \eta^{(i)}(y)$ we have $\nabla_\varsigma \log p(y) = \frac{\sum_{i=1}^{N} \nabla_\varsigma w^{(i)} \cdot \eta^{(i)}(y)}{\mathbf{w}^\top \boldsymbol{\eta}(y)}$, so that we can focus on the gradient in the enumerator. For a non-parametric density, $\eta^{(i)}(y) = |\mathbf{B}^{(i)}|^{-1}\kappa(\widehat{y}^{(i)})$, where we have used the shorthand $\widehat{y}^{(i)} := \mathbf{B}^{(-i)}(y - \mu^{(i)})$.

**Kernel Weights.** Eq. (13) follows directly from $\sum_{i=1}^{N} \nabla_\mathbf{w} w^{(i)} \cdot \eta^{(i)}(y) = \sum_{i=1}^{N} e_i \cdot \eta^{(i)}(y) = \boldsymbol{\eta}(y)$, where $e_i$ is the $i$-th standard basis vector.

**Kernel Centers.** The partial derivative *w.r.t.* the $k$-th kernel center $\mu^{(k)}$ is zero unless $k = i$, in which case

$$
\begin{aligned}
\frac{\partial\, w^{(i)}\eta^{(i)}(y)}{\partial\, \mu^{(i)}} &= w^{(i)}|\mathbf{B}^{(i)}|^{-1}\frac{\kappa(\widehat{y}^{(i)})}{\partial\mu^{(i)}} \\
&= w^{(i)}|\mathbf{B}^{(i)}|^{-1}\left(-\mathbf{B}^{(-\top i)}\right)\frac{\partial\,\kappa(\widehat{y}^{(i)})}{\partial\,\widehat{y}^{(i)}} \\
&= -w^{(i)}\frac{\mathbf{B}^{(-\top i)}}{|\mathbf{B}^{(i)}|}\frac{\partial\,\kappa(\widehat{y}^{(i)})}{\partial\,\widehat{y}^{(i)}},
\end{aligned}
\tag{26}
$$

which is the $i$-th component of the gradient $\nabla_{\boldsymbol{\mu}} \sum_{i=1}^{N} w^{(i)} \cdot \eta^{(i)}(y)$.

**Kernel Bandwidths.** Using the identity $\frac{\partial\, |\mathbf{B}|}{\partial\, \mathbf{B}} = |\mathbf{B}| \cdot \mathbf{B}^{-\top}$, we obtain

$$
\frac{\partial\, |\mathbf{B}|^{-1}}{\partial\, \mathbf{B}} = -\mathbf{B}^{-\top} \cdot |\mathbf{B}|^{-1}
\tag{27}
$$

and thus for the partial derivative *w.r.t.* the $i$-th kernel bandwidth matrix (all others being zero again)

$$
\begin{aligned}
\frac{\partial\, w^{(i)}\eta^{(i)}(y)}{\partial\, \mathbf{B}^{(i)}} &= w^{(i)}\left(\frac{\partial\, |\mathbf{B}^{(i)}|^{-1}}{\partial\, \mathbf{B}^{(i)}}\kappa(\widehat{y}^{(i)}) + |\mathbf{B}^{(i)}|^{-1} \cdot \frac{\partial\,\kappa(\widehat{y}^{(i)})}{\partial\, \mathbf{B}^{(i)}}\right) \\
&= w^{(i)}\left(-\mathbf{B}^{(-\top i)} \cdot |\mathbf{B}^{(i)}|^{-1}\kappa(\widehat{y}^{(i)}) + |\mathbf{B}^{(i)}|^{-1} \cdot \frac{\partial\,\kappa(\widehat{y}^{(i)})}{\partial\, \mathbf{B}^{(i)}}\right) \\
&= w^{(i)}|\mathbf{B}^{(i)}|^{-1}\left(-\mathbf{B}^{(-\top i)} \cdot \kappa(\widehat{y}^{(i)}) + \frac{\partial\,\kappa(\widehat{y}^{(i)})}{\partial\, \mathbf{B}^{(i)}}\right),
\end{aligned}
\tag{28}
$$

where the derivative of the kernel *w.r.t.* the bandwidth is

$$
\frac{\partial\,\kappa(\widehat{y}^{(i)})}{\partial\, \mathbf{B}^{(i)}} = -\mathbf{B}^{(-\top i)}\frac{\partial\kappa(\widehat{y}^{(i)})}{\partial\widehat{y}^{(i)}}(y - \mu^{(i)})^\top \mathbf{B}^{(-\top i)},
\tag{29}
$$

resulting in

$$
\frac{\partial\, w^{(i)}\eta^{(i)}(y)}{\partial\, \mathbf{B}^{(i)}} = -w^{(i)}\frac{\mathbf{B}^{(-\top i)}}{|\mathbf{B}^{(i)}|}\left(\kappa(\widehat{y}^{(i)}) + \frac{\partial\kappa(\widehat{y}^{(i)})}{\partial\widehat{y}^{(i)}}\widehat{y}^{(\top i)}\right).
\tag{30}
$$

### A.2.1 Backward-pass with Gaussian Kernels

The backward-pass through non-parametric conditional distributions with isotropic Gaussian kernels can be obtained from the previous section by plugging in the corresponding Gaussian densities and gradients. We introduce the recurring term $l^{(i)} := w^{(i)} \mathcal{N}(y \mid \mu^{(i)}, \mathbf{B}^{(2i)})$ as a shorthand and note that in the Gaussian case $\frac{\partial \, \kappa(\widehat{y}^{(i)})}{\partial \, \widehat{y}^{(i)}} = -\kappa(\widehat{y}^{(i)}) \cdot \widehat{y}^{(i)}$.

**Kernel Weights.**   Plugging in the Gaussian density directly leads to

$$\frac{\partial \, w^{(i)} \eta^{(i)}(y)}{\partial \, w^{(i)}} = l^{(i)} w^{(-i)}. \tag{31}$$

**Kernel Centers.**   Using the derivative of a Gaussian density stated above gives

$$\frac{\partial \, w^{(i)} \eta^{(i)}(y)}{\partial \, \mu^{(i)}} = l^{(i)} \mathbf{B}^{(-\top i)} \widehat{y}^{(i)}. \tag{32}$$

**Kernel Bandwidths.**   Another straightforward application of the derivative of a Gaussian density results in

$$\frac{\partial \, w^{(i)} \eta^{(i)}(y)}{\partial \, \mathbf{B}^{(i)}} = l^{(i)} \big( -\mathbf{B}^{(-\top i)} + \mathbf{B}^{(-\top i)} \widehat{y}^{(i)} \widehat{y}^{(\top i)} \big). \tag{33}$$

### A.3  Learning Non-parametric Structured Output Networks on Image Patches

Following the data extraction and preparation protocol described in Section 5.1, we generate a dataset consisting of $10 \times 10$ image patches. The two main challenges of this setup are the large number of nodes and the lack of a canonical graph structure. We approach the latter by computing the maximum spanning tree of a fully-connected graph whose edge weights are given by the continuous mutual information between the corresponding nodes [41]. The inferred graph structure (Figure 3b) validates the intuition that neighboring pixels carry most information about each other and should thus be modeled together. Using this optimal arborescence, we train a non-parametric structured output network according to Section 3 and evaluate the resulting model in the same way as we did in Table 2a. Our results are shown in Table 3a and confirm that the inferred dependencies are meaningful and allow us to take advantage of our structured representation. As before, we obtain a more accurate model of the true posterior than the two parametric approaches and the non-parametric approach without an explicit dependence structure.

| | Model | Non-param. | Structured | Parameter Group Estimation | | | |
| | | | | $-W$ | | $+W$ | |
| | | | | $-B$ | $+B$ | $-B$ | $+B$ |
|---|---|---|---|---|---|---|---|
| Neural Network + | Gaussian | ✗ | ✗ | $-8.46$ | $-13.75$ | $-9.16$ | $-19.03$ |
| | GGM [39] | ✗ | ✓ | $-8.69$ | $+17.78$ | $-9.05$ | $+18.60$ |
| | Mixture Density [40] | ✓ | ✗ | $+31.91$ | $+77.28$ | $+57.16$ | $+76.34$ |
| | NGM-100 (**ours**) | ✓ | ✓ | $+131.15$ | $+138.66$ | $+143.59$ | $+\mathbf{153.03}$ |

(a) Ablation Study

(b) Inferred Graph Structure

Table 3: **Quantitative Evaluation on Image Patches. (a)** We report the expected log-likelihood under the predicted posterior $p_{\boldsymbol{\theta}(\mathbf{x})}(\mathbf{y})$ on a test set consisting of corrupted $10 \times 10$ image patches. The conclusions of Section 5.2 generalize from 1D pixel stripes to 2D image patches, confirming once again the advantages of a structured and non-parametric approach. **(b)** The optimal arborescence is inferred automatically and connects neighboring pixels in the same row. Dependencies across rows are weaker, and thus sparse, but cannot be ignored completely due to the constraint of a tree-structured topology.

## Footnotes

[4]We write $\mathbf{B}^{(-i)} := \left(\mathbf{B}^{(i)}\right)^{-1}$ and $\mathbf{B}^{-\top} := \left(\mathbf{B}^{-1}\right)^\top = \left(\mathbf{B}^\top\right)^{-1}$ to avoid double superscripts and $\mathbf{B}^{(2i)} := \mathbf{B}^{(i)} \left(\mathbf{B}^{(i)}\right)^\top = \left(\mathbf{B}^{(i)}\right)^\top \mathbf{B}^{(i)}$ for the Gramian.