[Reviews · NeurIPS 2017]

Reviewer 1



The paper proposes Non-parametric Neural Networks (N3) a method that combines advantages of deep models for learning strong relations between input and output variables with the capabilities of probabilistic graphical models at modeling relationships between the output variables. Towards this goal, the proposed method is designed based on three components: a) a deep neural network (DNN) which learns the parameters of local non-parametric distributions conditioned on the input variables, b) a non-parametric graphical model (NGM) which defines a graph structure on the local distributions considered by the DNN. and, c) a recurrent inference network (RIN), which is trained/used to perform inference on the structured space defined by the DNN+NGM. The proposed method is sound, well motivated and each of its components are properly presented. The method is evaluated covering a good set of baselines and an ablation study showing variants of the proposed method. The evaluation shows that state of the art results are achieved by the proposed method. However, due to its technical nature, there is a large number of variables and terms that complicate the reading at times. Still this is just a minor point. Overall all the paper proposes a method with good potential in a variety of fields. However, one of my main concern is that its evaluation is conducted in the self-defined task of analyzing the statistics of natural pixels from the MS COCO dataset. In this regard, I would recommend adding more details regarding the evaluation, especially on how performance is measured for that task. Moreover, given that the evaluation is related with visual data, I would highly recommend evaluating their method on a more standard task from computer vision. In addition, very recently a paper with the very same title as this manuscript ("non-parametric neural networks") came out, i.e. Phillip and Carbonell, ICLR'17, while the method and goal approached in this paper is different from that from ICLR, I would strongly suggest either, a) make an explicit and strong positioning of your paper w.r.t. Phillip and Carbonell, ICLR'17, or b) consider an alternative (or extended) title for your paper. In this regard, I consider that your paper is better suited for the N3 title, however, this is a conflict that should be addressed adequately. Finally, given the potential of the proposed method I would encourage the authors to release the code related to the experiments presented in the manuscript. Are there any plans for releasing the code? I believe, this should encourage the adoption of the proposed method by the research community. I would appreciate if the authors address the last three paragraphs in their rebuttal.

Reviewer 2



The paper proposes a novel way to build a non-parametric graphical model with continuous variables using deep neural networks. Deep network is trained to predict parameters of local non-parametric distributions. Then the distributions are transformed to conditional distributions and recurrent network perform message passing-like procedure to infer required statistics. The paper is clearly written and the proposed approach seems novel and elegant to me. My main concern is the experimental evaluation in the paper. The authors use only small synthetic data. While the performed comparison on the dataset gives some insights, I strongly believe, that real world experiments would make the paper potentially very impactful.

Reviewer 3



It's my impression that an article should be self-contained. In the case of your article, it seems that a number of central details have been relegated to appendices. The lack of detail concerning how you condition via the conditioning operator is particularly problematic. 1) how the inference network is run and trained is less than clear. I understand you're doing approximate neural message passing via an RNN, but do you merely pass messages once or does the setup allow for loopy belief propagation? If yes, when does it terminate? 2) continuing the above line of inquiry, I assume you have to train a separate RIN for each statistic. If yes, how is the inference loss L_I selected? The loss you picked for marginal inference seems appropriate, but it's unclear what has to change for other statistics. 3) are you actually enforcing structure in your experiement? It goes unmentioned. I assume structure is decided a priori, unless its inference is a detail hidden away in the appendix? 4) I am not convinced that the way you use BB-VI in the experiments makes for a fair comparison. One, it's hardly the gold standard for variational inference given its instability problems. In your case you should at the very least be able to get away with reparameterized gradient estimators. Also, the advantage of BB is that you can apply it to any q-model while you use a notoriously weak mean-field approximation. Aside from the above, I enjoyed the article and its perspectives.